# Simulated patient training to improve youth engagement in HIV care in Kenya: A stepped wedge cluster randomized controlled trial

**Pamela K. Kohler**[1,2]*, **Cyrus Mugo**[3,4], **Kate S. Wilson**[1], **Hellen Moraa**[3,5],
**Alvin Onyango**[3,5], **Kenneth Tapia**[1], **Kenneth Pike**[2], **Caren Mburu**[1], **Margaret Nduati**[3],
**Brandon Guthrie**[1,4], **Barbra A. Richardson**[1,6,7], **Tamara Owens**[8], **David Bukusi**[3,5],
**Irene Inwani**[5], **Grace John-Stewart**[1,4,9], **Dalton Wamalwa**[3]

1 Department of Global Health, University of Washington, Seattle, Washington, United States of America,
2 Department of Child, Family, and Population Health Nursing, Seattle, Washington, United States of America, 3 Department of Pediatrics and Child Health, University of Nairobi, Nairobi, Kenya, 4 Department of Epidemiology, University of Washington, Seattle, Washington, United States of America, 5 Kenyatta National Hospital, Nairobi, Kenya, 6 Department of Biostatistics, University of Washington, Seattle, Washington, United States of America, 7 Division of Vaccine and Infectious Disease, Fred Hutch Cancer Center, Seattle, Washington, United States of America, 8 Howard University, Washington, DC, United States of America, 9 Department of Pediatrics, University of Washington, Seattle, Washington, United States of America

* pkohler2@uw.edu

**Data Availability Statement:** Medical record data are owned by the Kenyan government. Contact the University of Washington Office of Nursing

## Abstract

Youth living with HIV (YLHIV) report that negative interactions with health care workers (HCWs) affects willingness to return to care. This stepped wedge randomized trial evaluated effectiveness of a standardized patient actor (SP) HCW training intervention on adolescent engagement in care in Kenya. HCWs caring for YLHIV at 24 clinics received training on adolescent care, values clarification, communication, and motivational interviewing, with 7 SP encounters followed by facilitated feedback of videotaped interactions. Facilities were randomized to timing of the intervention. The primary outcome was defined as return within 3 months after first visit (engagement) among YLHIV who were either newly enrolled or who returned to care after >3 months out of care. Visit data was abstracted from electronic medical records. Generalized linear mixed models adjusted for time, being newly enrolled, and clustering by facility. YLHIV were surveyed regarding satisfaction with care. Overall, 139 HCWs were trained, and medical records were abstracted for 4,595 YLHIV. Median YLHIV age was 21 (IQR 19–23); 82% were female, 77% were newly enrolled in care, and 75% returned within 3 months. Half (54%) of trained HCWs remained at their clinics 9 months post-training. YLHIV engagement improved over time (global Wald test, p = 0.10). In adjusted models, the intervention showed no significant effect on engagement [adjusted Prevalence Ratio (aPR) = 0.95, 95% Confidence Interval (CI): 0.88–1.02]. Newly enrolled YLHIV had significantly higher engagement than those with prior lapses in care (aPR = 1.18, 95%CI: 1.05–1.33). Continuous satisfaction with care scores were significantly higher by wave 3 compared to baseline (coefficient = 0.38, 95%CI: 0.19–0.58). Despite provider skill improvement, there was no effect of SP training on YLHIV engagement in care. This may be due to temporal improvements or turnover of trained HCWs. Strategies to retain SP-training benefits need to address HCW turnover. YLHIV with prior gaps in care may need

Research for assistance with data requests: onrhelp@uw.edu.

**Funding:** This work was supported by the Eunice Kennedy Shriver National Institute of Child Health and Human Development (NICHD) (R01 HD085807 to PK). Statistical support was provided by the UW/FHCRC Center for AIDS Research (CFAR) (P30 AI027757). The funders had no role in study design, data collection and analysis, decision to publish, or preparation of the manuscript.

**Competing interests:** The authors have declared that no competing interests exist.

more intensive support. Registration CT #: NCT02928900. https://clinicaltrials.gov/ct2/show/NCT02928900.

## Introduction

Adolescents and young adults in Eastern and Southern Africa make up 70% of youth living with HIV (YLHIV) worldwide [1]. In 2020, antiretroviral coverage of YLHIV ages 10–19 in Kenya was 74% [1], falling short of 95-95-95 goals. YLHIV also have poor care outcomes compared to other age groups, with consistently higher rates of loss to follow up and lower rates of viral suppression than adults or young children [2–4]. In Kenya, a 2012–2016 review of national data found elevated viral load among adolescents with HIV 10–19 years was double that of adults (37% vs 13%) [5]. Globally, adolescent HIV-related deaths have declined just 37% since 2010, while child mortality has declined by 60% [6].

Adolescence is a time of rapid physiological, developmental, and psychosocial changes, characterized by increasing autonomy and identity formation [7]. These unique changes may aggravate preexisting stressors associated with HIV infection, increase vulnerability to depression and mental health concerns, and exacerbate poor adherence and retention in care [8–11]. Recognition of the distinct needs of adolescents within HIV service delivery has prompted global initiatives to promote "adolescent friendly" health services (AFHS) [12,13]. The World Health Organization recommends that AFHS are not simply separate infrastructure and space, but also a change in health provider attitudes and training. Adolescent friendly care aims to ensure that health providers are "non-judgmental and considerate in their dealings with adolescents" and that they have the "competencies to deliver the right health services in the right way" [12]. The Kenyan Ministry of Health adopted these recommendations, revising 2005 guidance in youth friendly health services in 2016 [14].

While health providers report recognition of adolescent needs and awareness of adolescent friendly guidelines, many also still report lack of confidence in how to engage with and care for adolescents. A study of reproductive health providers in Kenya documented a disconnect between advertised "youth friendly services" and providers' lack of competency and training to carry out adolescent friendly care, especially related to counselling and interpersonal communication [15]. They describe feeling conflicted between their own personal feelings, cultural and religious values and beliefs, and their wish to support young people's rights to accessing services. In another study, health care workers tasked with providing adolescent HIV testing services reported feeling inadequately prepared to cope with the needs of this age group, lack of confidence in counseling skills, and fear of doing the wrong thing [16].

Standardized patient (SP) training presents a unique opportunity to train health care personnel to effectively deliver AFHS. SPs are trained actors that work with health providers in mock clinical encounters for the purposes of training and evaluation [17]. Used since the 1960s in clinical and medical education and for licensing exams in the United States and Europe [18], SPs have been less widely used in resource-limited settings. SP training has been shown both to improve clinical performance including provider skills in empathy, patient-centered communication skills, and counseling [19,20] as well as patient outcomes [21–23].

This study aimed to develop and evaluate a clinical training intervention utilizing standardized patient actors to improve communication and interpersonal skills of health care workers in working with adolescents, resulting in increased engagement in HIV care. Our hypothesis, informed by Andersen's Model of Healthcare Utilization [24,25], was that a SP training program would improve the health care environment and patient-centered HIV services through

provider factors of building trust and empathy, enhancing communication skills, and increasing provider clinical and counseling knowledge. This improvement in delivery of care could, in turn, address barriers cited by adolescents and improve linkage to, and retention in, vital HIV services.

## Methods

### Ethics statement

This study was reviewed and approved by the University of Washington (51926) institutional review board and the University of Nairobi—Kenyatta National Hospital (P476/06/2016) ethical review committee. Health worker participants provided written informed consent. Adolescent surveys were anonymous, thus we received approval to waive written consent and allow adolescents 14 and older who attended care independently to provide oral self-consent. Adolescents under the age of 14 provided oral assent and their caregiver provided oral consent. De-identified medical records were provided through an agreement with the Ministry of Health.

### Design

A stepped wedge cluster randomized controlled trial (RCT) was conducted to evaluate a clinical training intervention using standardized patient actors to improve communication and interpersonal skills of health care workers caring for adolescents and young adults, as a means to increase YLHIV engagement in care [26]. Twenty-four health care facilities (clusters) were randomized to intervention sequence (Fig 1, S1 Checklist).

The stepped wedge study design was selected because intervention-delivery occurred at the clinic, rather than individual level, to facilitate cluster recruitment, and for logistical considerations as the training was not feasible to implement simultaneously at a large number of clinics [27]. This design also can have greater power than a parallel design when there are substantial intra-cluster correlations (i.e. within facilities) or large numbers of clusters [28]. Randomization at a clinic level to *when the intervention is introduced* allowed for the traditional benefits of randomization, while allowing the minimal risk intervention to be provided to all sites. Training was conducted with health providers from participating facilities at 4 time points, 9 months apart (Table 1). Outcomes were assessed among YLHIV in participating facilities over a period of 15 months following each training. The final 6 months of outcome assessment was truncated due to extensive changes in patient visit schedules due to the COVID-19 pandemic.

### Population and setting

Eligible clinics had more than 40 adolescents, had a current electronic medical record (EMR) system, and no other special adolescent interventions. From 24 enrolled facilities, up to ten health providers from each facility were selected for the training. Eligible providers included clinical officers, doctors, nurses, and counselors who provide direct clinical services to adolescents. Providers were identified by the study coordinator in collaboration with the facility manager. Turnover of health providers at their study sites was monitored during the study period and newly hired providers enrolled if sites had not yet participated in the intervention. Retention in care and clinical outcomes were assessed via an audit of EMR charts from YLHIV 10–24 years of age, and a subset of YLHIV from each facility were also enrolled to complete patient satisfaction and risk behavior surveys after each training wave.

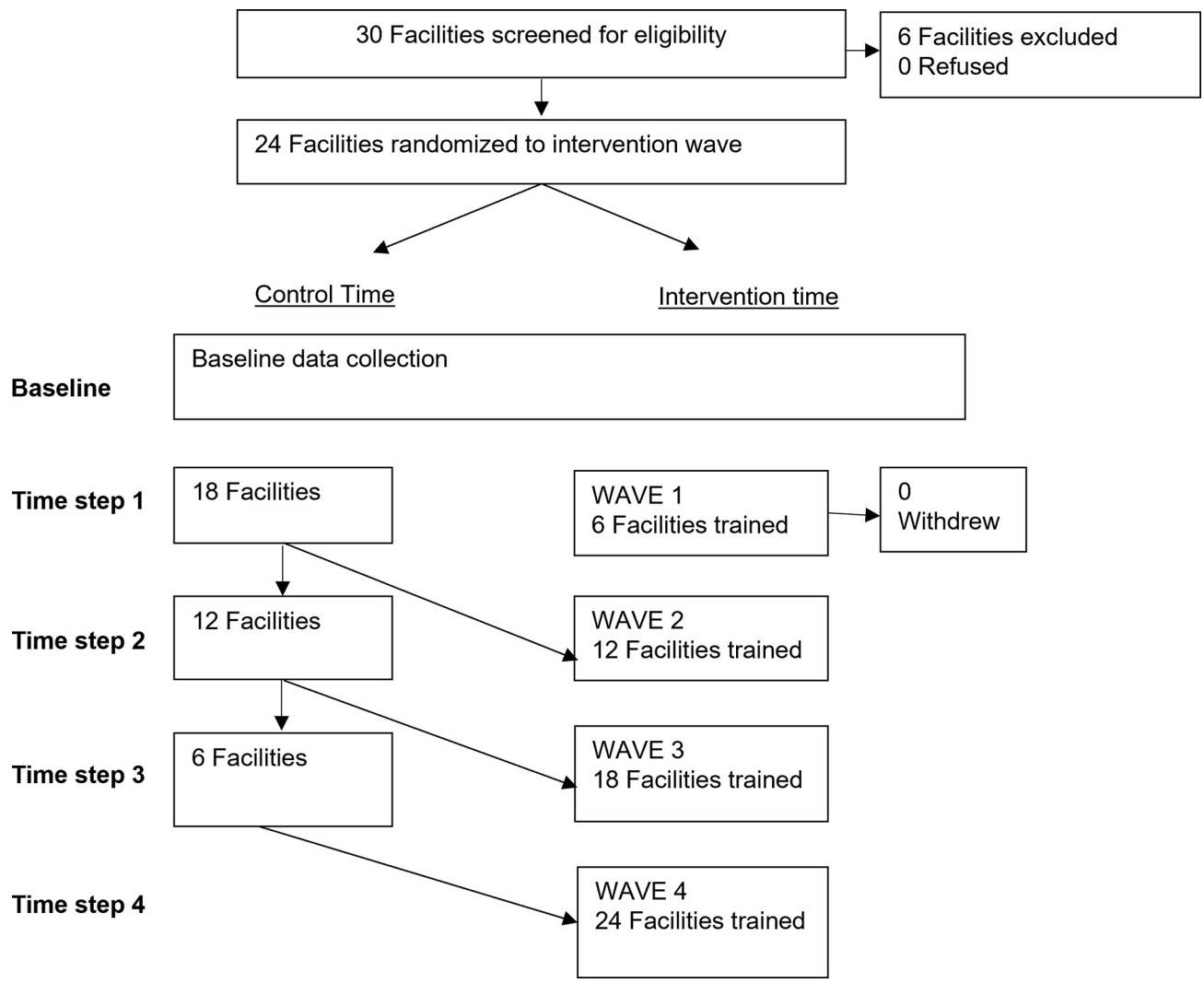

**Fig 1. Adapted CONSORT diagram of SPEED intervention.**

## Intervention

SP training is anchored in Kolb's experiential learning and Bandura's social learning theories [29,30]; health providers learn through cycles of concrete experiences (mock clinical encounters with SPs), observation, and feedback after the encounters, in a safe and controlled setting. Our 2-day training program for adolescent HIV care providers consisted of a combination of

**Table 1. Overview of stepped-wedge trial timing, by cluster wave.**

| Clinic clusters | Introduction of Intervention | | | | |
|---|---|---|---|---|---|
| | Baseline | Month 1 | Month 10 | Month 19 | Month 28 |
| Wave 1: Sites 1–6 | | | | | |
| Wave 2: Sites 7–12 | | | | | |
| Wave 3: Sites 13–18 | | | | | |
| Wave 4: Sites 19–24 | | | | | |

didactic sessions and SP encounters. Lectures included content in adolescent development and national guidelines for YLHIV care, communication skills, and motivational interviewing. Group exercises in communication and values clarification were also conducted. Participant providers then rotated through 7 different SP encounters, followed by individual feedback from SPs and a facilitated group debriefing in which the training group reviewed a video recording of one of each participants' SP encounters. Lectures and facilitation were delivered by study team clinicians with expertise in adolescent HIV care.

Instrument development and actor training followed the Association of Standardized Patient Educators (ASPE) Standards of Best Practice [31]. Cases were based on literature review, previous qualitative work [16,32], and an expert advisory panel. The seven cases included a practice scenario with no treatment challenges, advancing in difficulty and including issues of disclosure, adherence, sexually transmitted infections, gender-based violence, family planning, alcohol and drug use, depression, and sexual identity [32]. Actors were recruited and auditioned through a casting agency and trained in the SP methodology and the specific cases by an expert trainer and study staff.

The intervention training schedule was planned to minimize disruption to clinic operations and accommodate provider availability from April 2017 through October 2019. During each wave it took 2–4 weeks to train in groups of 6 providers. Participants from different clinics in the Nairobi and Kiambu counties and those in Western Kenya sites trained together. Once clinic staff were trained, it was presumed that all YLHIV enrolled in care at their respective facilities were exposed to the intervention.

## Outcomes

The primary outcome for this study was youth retention in HIV care, specifically during early engagement (Table 2). Engagement was defined as returning for a first follow-up visit within 3 months of enrollment. To ensure adequate sample, YLHIV who had previously been lost to follow-up (LTFU) were included with an engagement definition of return for a follow-up visit within 3 months after re-engagement in care. Previous LTFU was defined as no clinic record for at least 3 months and no record of death or transfer to another facility. Hypothesized process measures intended to clarify the causal pathway included provider competency ratings before and after the intervention and adolescent satisfaction with care. Satisfaction with care questions were derived from concepts listed in the Kalamazoo consensus [33] and adapted from a validated youth-provider interaction assessment [34]. Overall satisfaction was recorded on a Likert scale (very unsatisfied = 1, somewhat unsatisfied = 2, neutral = 3, somewhat

**Table 2. Study outcomes, definitions, and sources.**

| Primary Outcome | Definition | Source |
|---|---|---|
| Retention in care | Return to clinic within 90 days of enrollment visit, or within 30 days of a re-engagement visit after a lapse of 90 days | EMR |
| **Process Outcomes** | | |
| HCW competency | Self-reported competency before and after training | Provider Survey |
| Satisfaction with care | Mean youth satisfaction score per clinic | YLHIV survey |
| **Secondary Outcomes** | | |
| Retention among all ALHIV | Return to clinic within 30 days of enrollment visit | EMR |
| Longer retention window | Return to clinic within 180 days and 365 days | EMR |

satisfied = 4, very satisfied = 5). Satisfaction surveys were completed at baseline and after waves 1–3 of training. Wave 4 was not completed as a result of the COVID-19 pandemic. Finally, secondary measures of retention within longer time frames (180 days and 365 days) or among all adolescents currently enrolled in HIV care were evaluated.

## Sample size

The methods described by Hussey and Hughes [35] based on a model with a random intercept were used to determine power assuming an evaluation with 24 clusters and 5 time points (including baseline). Twenty-four clinics were included to account for a possible 15% drop-out without replacement. As a small proportion of the adolescents enrolled in HIV care at each site would be newly enrolled or recently re-engaged in care, the minimum number of records per clinic was estimated for 80% power to detect a 15% difference between control and intervention periods (i.e. 75% versus 90%) assuming a coefficient of variation k of 0.25 and calculated power for a two-tailed test with $\alpha = 0.05$. Based on relevant published data [2], 75% of adolescents in the control period were assumed to return to clinic after first visit. An estimated minimum of 5 adolescents per clinic per time point were needed under these assumptions.

## Randomization and sequence generation

Facility clusters were randomized to the sequence in which they received the intervention. In this one-way cross-over design, all 24 sites eventually received the intervention. Groups of 6 clinics were randomized to one of four intervention waves using stratified randomization according to region and by facility size. Facility size was defined as 'high volume', more than 73 YLHIV enrolled and medium volume, 73 or fewer YLHIV enrolled, based on the median, to ensure balance of characteristics in each wave [36]. The study statistician generated the randomization assignment for each clinic using Microsoft Excel. Study staff were not blinded to randomization so that they could schedule and implement the training. Facilities were not informed which wave they were in until it was time to schedule that wave.

## Statistical methods

To estimate the effect of the intervention on the individual level, a generalized linear mixed regression model with a Poisson distribution and robust standard errors, allowing for random effects for clusters and fixed effects for time was used. Adjusted prevalence ratios (aPR) and 95% confidence intervals at the 5% significance level (two-sided) for primary outcomes were estimated. Satisfaction with care used the same approach with a Gaussian distribution for linear outcomes and adjusted for sex and age category (10–14, 15–19, 20–24). Analyses were conducted using Stata 17 (College Station, TX).

# Results

## Recruitment and enrollment

Overall, 227 providers were screened for eligibility, 100% consented to participate, and 139 participated in the intervention training. At baseline, 157 providers were enrolled, and new providers continued to be consented at enrolled sites immediately before their respective training wave. Medical records covering July 2016—March 2020 were abstracted for 4,595 YLHIV with 44,622 visits. Although YLHIV EMR data through December 2020 was planned to be included, outcome monitoring was stopped in March 2020 due to the influence of the COVID-19 pandemic on clinic attendance.

## Baseline characteristics

Among the 24 participating facilities, 9 (38%) were in Nairobi, 7 (29%) in Kiambu, 4 (17%) in Kisumu, and 4 (17%) in Homa Bay county. Most (n = 10, 42%) were sub-county level hospitals, with 5 (21%) county-level hospitals or higher, and 8 (33%) health centers (Table 3). At baseline, facilities reported a median number of 1,707 clients enrolled in HIV care, including a median of 83 YLHIV. Prior to start of study activities, 5 facilities (21%) had participated in any previous AFHS training.

Among 139 HCWs completing the training, 135 participants completed demographic surveys. They were a median age of 33 years and 72% female. Most (42%) were counselors, clinical officers (28%), and nurses (26%). HCWs reported a median of 3 years of experience caring for YLHIV (IQR = 1–5) and half (50%) had received any prior training in AFHS (S1 Data).

Among 6,043 YLHIV in care during the study time frame, 1,448 who were continuing care were excluded and the primary analysis was restricted to 4,595 YLHIV: 3,540 newly enrolled and 1,055 recently returned clients (Table 4). The median number of YLHIV per facility cluster was 218 (IQR = 107–243). Most YLHIV (65%) were age 20–24, 24% were 15–19, and 11% were 10–14; 82% were female. Overall, 75% of newly enrolled and recently returned YLHIV

**Table 3. Baseline characteristics of participants.**

| Characteristic | N (%) or Median (IQR) |
| --- | --- |
| **Facilities (N = 24)** | |
| Region | |
| Nairobi | 9 (37.5) |
| Kiambu | 7 (29.2) |
| Kisumu | 4 (16.7) |
| Homa Bay | 4 (16.7) |
| Facility type | |
| County referral hospital or higher | 6 (25.0) |
| Sub-county hospital | 10 (41.7) |
| Health center | 8 (33.3) |
| Patient volume – adults | 1,707 (921–2327) |
| Patient volume – adolescents (10–24 years) | 83 (40–153) |
| Any adolescent friendly health services training | 5 (20.8) |
| HCW Cadres on staff | |
| Clinical Officer | 3 (2–5) |
| Nurse | 5 (2–11.0) |
| HTC Counselor | 5 (2–5.5) |
| **HCW participants (N = 139)** | |
| Age | 33 (24–40) |
| Gender | |
| Male | 33 (24.4) |
| Female | 97 (71.9) |
| Not reported | 5 (3.7) |
| Position | |
| Counselor | 57 (42.2) |
| Nurse | 35 (25.9) |
| Clinical Officer | 38 (28.2) |
| Not reported | 5 (3.7) |
| Years working with YLHIV | 3 (1–5) |
| Any prior training in adolescent-friendly services | 68 (50.4) |

**Table 4. Adolescent EMR records (n = 4,595).**

| Characteristic | Overall<br>N (%) or Median (IQR) | Intervention exposed<br>N = 2,123 | Intervention unexposed<br>N = 2,472 |
|---|---|---|---|
| Gender | | | |
| Female | 3,782 (82.3) | 1,761 (83.0) | 2,021 (81.8) |
| Male | 812 (17.7) | 362 (17.0) | 450 (18.2) |
| | | | |
| Age | | | |
| 10–14 | 498 (10.8) | 220 (10.4) | 278 (11.3) |
| 15–19 | 1,113 (24.2) | 474 (22.3) | 639 (25.9) |
| 20–24 | 2,984 (64.9) | 1,429 (67.3) | 1,555 (62.9) |
| Age (median years, IQR) | 21.1 (18.9–22.7) | 21.3 (19.1–22.8) | 20.9 (18.7–22.6) |
| Marital status | | | |
| Married | 1,174 (25.6) | 529 (24.9) | 645 (26.1) |
| Not married | 1,517 (33.0) | 670 (31.6) | 847 (34.3) |
| Widowed | 15 (0.3) | 2 (0.1) | 13 (0.5) |
| Not reported | 1,889 (41.1) | 922 (43.4) | 967 (39.1) |
| Enrollment in care | | | |
| New | 3,540 (77.0) | 1,504 (70.8) | 2,036 (82.4) |
| Recently returned | 1,055 (23.0) | 619 (29.2) | 436 (17.7) |
| First eligible visit during study wave | | | |
| Baseline | 899 (19.6) | 0 (0.0) | 899 (36.4) |
| Wave 1 | 1,049 (22.8) | 251 (11.8) | 798 (32.3) |
| Wave 2 | 1,064 (23.2) | 588 (27.7) | 476 (19.3) |
| Wave 3 | 1,350 (29.4) | 1,051 (49.5) | 299 (12.1) |
| Wave 4 | 233 (5.1) | 233 (11.0) | 0 (0.0) |
| Returned within 3 months after first visit | 3,425 (74.5) | 1,570 (74.0) | 1,855 (75.0) |

returned for a second visit within 90 days of their first visit on record (range across all facilities = 49.8% to 93.8%).

## Early engagement in care

Among the 4,595 adolescents and youth across 24 clusters, YLHIV presenting for care during the exposed period had similar probability of return for a second visit as those who presented during unexposed periods [74% in exposed, 75% in unexposed group; adjusted prevalence ratio (aPR) = 0.95, 95% confidence interval (CI) = 0.88–1.02] (Table 5). There was a trend for overall improved retention over time (global Wald p = 0.10) with significant improvement comparing baseline engagement to wave 2 (aPR = 1.08, 95%CI = 1.00–1.16) and to wave 3 (aPR = 1.15, 95%CI = 1.04–1.27). Newly enrolled clients had significantly higher engagement compared to those who had previous gaps in care across all time periods (aPR = 1.18, 95%CI = 1.05–1.33). Sensitivity analyses restricted to 10–19 year old YLHIV and among all YLHIV, including those continuing in care, were similarly null when comparing exposed and unexposed time periods (aPR = 0.91, 95%CI = 0.81–1.04 and aPR = 0.98, 95%CI = 0.90–1.07 respectively).

## Satisfaction with care

Across all 24 clusters, 925 YLHIV completed satisfaction with care exit surveys (S2 Data). YLHIV were a median of 16 years old, 58% female, and 73% enrolled in school. The majority of YLHIV presented to care alone (68%), 27% attended with a parent, and 5% with a friend or

**Table 5. Intervention exposure on early engagement (n = 4,595) and satisfaction with care (n = 925).**

| Primary Outcome – Early engagement in care | Prevalence ratio | 95% Confidence interval | p-value |
|---|---|---|---|
| Intervention exposure | 0.95 | 0.88–1.02 | 0.15 |
| Time | | | |
| Baseline | Ref | | |
| Wave 1 | 1.05 | 0.98–1.13 | 0.14 |
| Wave 2 | 1.08 | 1.00–1.16 | 0.04 |
| Wave 3 | 1.15 | 1.04–1.27 | 0.007 |
| Wave 4 | 1.14 | 0.99–1.31 | 0.06 |
| Newly enrolled | 1.18 | 1.05–1.33 | 0.006 |
| **Process Measure – Satisfaction with care** | **Coefficient** | **95% Confidence interval** | **p-value** |
| Intervention exposure | -0.06 | -0.19–0.07 | 0.37 |
| Time* | | | |
| Baseline | Ref | | |
| Wave 1 | 0.23** | 0.09–0.37 | 0.001 |
| Wave 2 | 0.29 | 0.10–0.48 | 0.003 |
| Wave 3 | 0.38 | 0.19–0.57 | <0.001 |
| Age | | | |
| 10–14 | Ref | | |
| 15–19 | 0.13 | 0.04–0.21 | 0.005 |
| 20–24 | 0.04 | -0.08–0.16 | 0.49 |
| Female | 0.86 | -0.04–0.11 | 0.11 |

*Wave 4 surveys not conducted due to COVID-19.

**Increase from baseline score on a 5-pt Likert scale.

other support person. YLHIV reported general overall satisfaction, with a median score of 4 at baseline at 5 at the end of wave 3. The largest differences were seen in the very satisfied category, with 47% reporting being very satisfied with care at baseline and 78% being very satisfied at the end of wave 3. Those presenting for care during exposed time periods had higher satisfaction (69.8% very satisfied) than those who were unexposed to trained providers (60.4% very satisfied), however in models adjusted for time, intervention exposure was not significantly associated with increased satisfaction (adjusted coefficient = -0.06, 95%CI = -0.19–0.07).

## Harms

The study team monitored for social harms that could have resulted from altered counseling approaches by providers following the training. None were reported.

## Health provider retention at study sites

Nine months after each training wave, we assessed whether trained providers were still practicing at their respective study sites. Overall, 54% (75/139) were still working at the same facility (facility-specific range: 20%-100%).

## Discussion

This stepped wedge randomized trial of a standardized patient training intervention was highly acceptable and improved provider confidence in care of YLHIV [37], however it did

not result in improved early engagement in care among YLHIV. Several measurement challenges limited ability to detect effectiveness on the YLHIV population including substantial temporal changes in study outcomes, truncated final wave ascertainment of YLHIV measures due to the COVID-19 pandemic, and turnover of trained HCWs resulting in diffusion of the intervention across sites.

Encouragingly, we found significant improvements over time in both retention and satisfaction with care outcomes, suggesting that Kenya's national investment in adolescent-friendly care is contributing to improved YLHIV outcomes. During the trial period Kenya's National AIDS and STD Control Programme launched a 2015 Adolescent Package of Care which aimed to minimize barriers and improve health service provision for adolescents [38,39], the Ministry of Health issued updated 2016 national guidance for adolescent friendly services [14], and PEPFAR sustained and prioritized investment in study counties and adolescent populations [40–42]. The subsequent 2018 Kenya Population-based HIV Impact Assessment (KENPHIA) found that adolescent (15–24 years) viral suppression was 79% compared to adult suppression at 72% [43]. It is possible that these national gains contributed to dilution of the overall intervention effect.

We also found that those with previous gaps in care were more likely to be lost to follow up again. This may have been in part because of longer visit intervals, though the study period pre-dated roll out of differentiated care and standardized longer visit intervals. It is also likely that adolescents have longer visit intervals related to school attendance. Previous gaps in care, however, could be a signal for health providers to allocate more intensive support services to prevent future loss to care.

At the individual level, YLHIV face numerous barriers to adherence and engagement in care, including depression, stigma, and other psychosocial factors [44–46]. While evidence suggests that investment in adolescent friendly services addresses many of these barriers and contributes to improved clinical outcomes among YLHIV [47–50], interventions for more challenging mental health conditions are still largely lacking for this population [51]. It is also possible that YLHIV in the intervention period no longer received exposure to the intervention due to the large degree of HCW turnover. Shifting training delivery to pre-service settings for all new HCWs may be useful to ensure consistent training coverage in priority topics and skills. In addition, evaluation of a more proximal measures (provider communication skills, judgmental behavior, or adherence to guidelines) or linking trained HCW with specific YLHIV in analyses rather than assessing clinic-level effect could enable a clearer understanding of intervention effect.

This study was unique in that we attempted to determine whether an SP training intervention had a long-term patient effect using a stepped wedge RCT design. While SPs are increasingly used to assess quality of care in low-resource settings [52–58], studies of patient actor training interventions have primarily taken place in high-resource settings with proximal outcomes of provider skill or performance [19,59,60]. Indeed, our intervention was effective in improving self-rated provider skills [32,37]. Our use of routine electronic medical records as a tool to assess patient outcomes was also innovative. Although a promising approach to measure trial and program outcomes, there were numerous challenges with completeness of EMR data and implementation of EMR systems. Next visit dates were not an established entry at the time of the study, thus assumptions had to be made about expected return dates. Similarly, disruptions in use of the 4 different electronic platforms across 6 implementing partners and 24 clinics, incompleteness of clinical variables such as viral load and adherence in the EMR.

## Conclusion

In conclusion, the standardized patient actor training intervention did not result in a measurable effect on adolescent satisfaction or early engagement in care among YLHIV in Kenya.

There was however high acceptability among health providers, and previous analyses demonstrated improvements in provider confidence and performance. Future research could target more proximal measures of patient-centered care as well as effectiveness for other key populations or other care systems. Training HCWs in their pre-service, rather than in-service, period may address facility turnover however may need to be reinforced over time with evolving guidelines and priorities.

## Supporting information

**S1 Checklist. Reporting checklist for stepped wedge cluster randomized trials (SW-CRT).**
(PDF)

**S1 Data. Health care worker survey data.** Health care worker codebook.
(CSV)

**S2 Data. Adolescent satisfaction survey data.** Adolescent satisfaction survey codebook.
(CSV)

## Author Contributions

**Conceptualization:** Pamela K. Kohler, Cyrus Mugo, Kate S. Wilson, Barbra A. Richardson, David Bukusi, Grace John-Stewart, Dalton Wamalwa.

**Data curation:** Alvin Onyango, Kenneth Tapia, Kenneth Pike, Margaret Nduati.

**Formal analysis:** Pamela K. Kohler, Kate S. Wilson, Kenneth Tapia, Kenneth Pike.

**Funding acquisition:** Pamela K. Kohler.

**Investigation:** Pamela K. Kohler, Cyrus Mugo, Hellen Moraa, Alvin Onyango, Caren Mburu, Margaret Nduati, David Bukusi, Irene Inwani, Dalton Wamalwa.

**Methodology:** Pamela K. Kohler, Kate S. Wilson, Brandon Guthrie, Barbra A. Richardson, Tamara Owens, Grace John-Stewart.

**Supervision:** Pamela K. Kohler, Irene Inwani, Dalton Wamalwa.

**Writing – original draft:** Pamela K. Kohler.

**Writing – review & editing:** Cyrus Mugo, Kate S. Wilson, Hellen Moraa, Alvin Onyango, Caren Mburu, Margaret Nduati, Brandon Guthrie, Barbra A. Richardson, Tamara Owens, David Bukusi, Irene Inwani, Grace John-Stewart, Dalton Wamalwa.

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
