## [Decision Letter · Decision Letter 0]

19 Dec 2022

PGPH-D-22-01415

Simulated patient training to improve youth engagement in HIV care in Kenya: a stepped wedge cluster randomized controlled trial

Dear Dr. Kohler,

Thank you for submitting your manuscript to PLOS Global Public Health. After careful consideration, we feel that it has merit but does not fully meet PLOS Global Public Health’s publication criteria as it currently stands. Therefore, we invite you to submit a revised version of the manuscript that addresses the points raised during the review process.

The reviewers find that the study methods presented in the current submission are sound and that the paper is well-written. However, they also mention several concerns which need to be addressed. For example, they feel that the methodology needs further clarification, e.g. more details should be provided regarding the inclusion and exclusion criteria for study sites and the power calculation performed for the study. In addition, they note that the results and discussion sections need revisions.

We look forward to receiving your revised manuscript.

Kind regards,

Alex Schaefer, PhD

Associate Editor

Journal Requirements:

2. Please send a completed 'Competing Interests' statement, including any COIs declared by your co-authors. If you have no competing interests to declare, please state "The authors have declared that no competing interests exist". Otherwise please declare all competing interests beginning with the statement "I have read the journal's policy and the authors of this manuscript have the following competing interests:"

3. Please amend your detailed Financial Disclosure statement. This is published with the article. It must therefore be completed in full sentences and contain the exact wording you wish to be published.

4. We have noticed that you have uploaded Supporting Information files, but you have not included a list of legends. Please add a full list of legends for your Supporting Information files after the references list.

Additional Editor Comments (if provided):

Reviewers' comments:

Reviewer's Responses to Questions

**Comments to the Author**

1. Does this manuscript meet PLOS Global Public Health’s publication criteria? Is the manuscript technically sound, and do the data support the conclusions? The manuscript must describe methodologically and ethically rigorous research with conclusions that are appropriately drawn based on the data presented.

Reviewer #1: Yes

Reviewer #2: Yes

2. Has the statistical analysis been performed appropriately and rigorously?

Reviewer #1: Yes

Reviewer #2: Yes

3. Have the authors made all data underlying the findings in their manuscript fully available (please refer to the Data Availability Statement at the start of the manuscript PDF file)?

Reviewer #1: Yes

Reviewer #2: Yes

4. Is the manuscript presented in an intelligible fashion and written in standard English?

Reviewer #1: Yes

Reviewer #2: Yes

5. Review Comments to the Author

Reviewer #1: Thank you for the opportunity to review this manuscript. The authors describe a stepped wedge cluster RCT evaluating the effectiveness of a standardized patient actor training on adolescent engagement in HIV care. The intervention is innovative and underscores important areas for future work. The study methods are sound and the paper is well-written, but several clarifications could improve its readability. I have suggested revisions and clarifications in the attached, organized by manuscript section.

Reviewer #2: The authors present the results of a cluster randomized stepped wedge trial of an intervention to improve heath care worker provision of youth friendly services in Kenya. The manuscript is very well written and clear. A few minor suggestions are provided below.

1) It was a little unclear whether the power was calculated for an increase of 15% from 75% (for an 86% intervention early engagement outcome) or a change from 75% to 90% (a 20% increase). I think it's latter.

2) It might be helpful to other researchers for the authors to also report the CAC and the IAC used in the power calculation.

3) The authors did a nice job of justifying stepped wedge over regular cluster randomized trial. Might be good for readers if you also list other advantages to the the stepped wedge other than lack of feasibility of initiating the intervention at a large number of clinics at the same time like minimally detectable difference, number of clusters needed, or the size of the clusters needed?

It's unfortunate that covid impacted ascertainment of outcomes in the final wave. The authors should be commended for reporting null findings so transparently.

6. PLOS authors have the option to publish the peer review history of their article (what does this mean?). If published, this will include your full peer review and any attached files.

**Do you want your identity to be public for this peer review?** For information about this choice, including consent withdrawal, please see our Privacy Policy.

Reviewer #1: No

Reviewer #2: No

---

## [Decision Letter · Decision Letter 1]

7 Mar 2023

Simulated patient training to improve youth engagement in HIV care in Kenya: a stepped wedge cluster randomized controlled trial

PGPH-D-22-01415R1

Dear Dr. Kohler,

We are pleased to inform you that your manuscript 'Simulated patient training to improve youth engagement in HIV care in Kenya: a stepped wedge cluster randomized controlled trial' has been provisionally accepted for publication in PLOS Global Public Health.

Best regards,

Julia Robinson

Executive Editor

Reviewer Comments (if any, and for reference):

Reviewer's Responses to Questions

**Comments to the Author**

1. If the authors have adequately addressed your comments raised in a previous round of review and you feel that this manuscript is now acceptable for publication, you may indicate that here to bypass the “Comments to the Author” section, enter your conflict of interest statement in the “Confidential to Editor” section, and submit your "Accept" recommendation.

Reviewer #2: All comments have been addressed

2. Does this manuscript meet PLOS Global Public Health’s publication criteria? Is the manuscript technically sound, and do the data support the conclusions? The manuscript must describe methodologically and ethically rigorous research with conclusions that are appropriately drawn based on the data presented.

Reviewer #2: Yes

3. Has the statistical analysis been performed appropriately and rigorously?

Reviewer #2: Yes

4. Have the authors made all data underlying the findings in their manuscript fully available (please refer to the Data Availability Statement at the start of the manuscript PDF file)?

Reviewer #2: Yes

5. Is the manuscript presented in an intelligible fashion and written in standard English?

Reviewer #2: Yes

6. Review Comments to the Author

Reviewer #2: The authors have addressed all of my comments. Thank you.

7. PLOS authors have the option to publish the peer review history of their article (what does this mean?). If published, this will include your full peer review and any attached files.

**Do you want your identity to be public for this peer review?** For information about this choice, including consent withdrawal, please see our Privacy Policy.

Reviewer #2: No
